Effects of acute levodopa challenge on resting cerebral blood flow in Parkinson’s Disease patients assessed using pseudo-continuous arterial spin labeling

Chen Yufen 1 yfchen@northwestern.edu
Pressman Peter 2
Simuni Tanya 3
Parrish Todd B. 1
Gitelman Darren R. 3 4 5
1 Department of Radiology, Feinberg School of Medicine, Northwestern University , Chicago, IL , USA
2 Department of Neurology, Memory and Aging Center, University of California , San Francisco, CA , USA
3 Department of Neurology, Feinberg School of Medicine, Northwestern University , Chicago, IL , USA
4 Department of Medicine, Advocate Lutheran General Hospital , Park Ridge, IL , USA
5 Department of Medicine, Rosalind Franklin University of Medicine and Science , North Chicago, IL , USA
Black Kevin
Electronic publication date: 2015 Nov 3
Publication date: 2015
Volume: 3
Electronic Location ID: e1381
Received 2015 Jun 17; Accepted 2015 Oct 14
Copyright: © 2015 Chen et al.
Copyright year: 2015
Copyright holder: Chen et al.
License: This is an open access article distributed under the terms of the Creative Commons Attribution License, which permits unrestricted use, distribution, reproduction and adaptation in any medium and for any purpose provided that it is properly attributed. For attribution, the original author(s), title, publication source (PeerJ) and either DOI or URL of the article must be cited.
License URL: https://creativecommons.org/licenses/by/4.0/

Keywords: Cerebral blood flow, Levodopa, Arterial spin labeling, Parkinson’s disease, Pharmacological MRI

Funding: Paul Ruby Foundation This study was supported by a grant from the Paul Ruby Foundation for Parkinson’s research. The funders had no role in study design, data collection and analysis, decision to publish, or preparation of the manuscript.

==============================
Introduction. Levodopa is the gold-standard for treatment of Parkinson’s disease (PD) related motor symptoms. In this study, we used pseudo-continuous arterial spin labeling (pCASL) to quantify changes in cerebral blood flow (CBF) after acute oral administration of levodopa in PD patients.

Materials and Methods. Thirteen patients (3 females, age 66.2 ± 8.7 years) with moderately advanced PD (Hoehn and Yahr stage >2 (median 2.5), disease duration >3 years) were scanned on a 3T Siemens MR scanner before and after oral levodopa administration. Statistical parametric mapping was used to detect drug-induced changes in CBF and its correlation to clinical severity scales. Images were normalized and flipped in order to examine effects on the more affected (left) and less affected (right) cerebral hemispheres across the cohort.

Results. Levodopa did not change global CBF but increased regional CBF in dorsal midbrain, precuneus/cuneus, more affected inferior frontal pars opercularis and triangularis, bilateral pre- and postcentral gyri, more affected inferior parietal areas, as well as less affected putamen/globus pallidus by 27–74% (p < 0.05, FWE corrected for multiple comparisons). CBF change was negatively correlated with improvement in bradykinesia UPDRS-III subscore in the more affected precentral gyrus, and total predrug UPDRS-III score in the mid-cingulate region. Drug-induced CBF change in a widespread network of regions including parietal and postcentral areas was also negatively correlated with the predrug rigidity UPDRS-III subscore.

Conclusion. These findings are in line with prior reports of abnormal activity in the nigrostriatal pathway of PD patients and demonstrate the feasibility of pCASL as a neuroimaging tool for investigating in vivo physiological effects of acute drug administration in PD.

Introduction

Parkinson’s disease (PD) is a neurodegenerative disorder where the classical pathophysiology is the loss of dopaminergic neurons in the substantia nigra and disruption of the nigrostriatal pathway (Stoessl, 2012), which results in motor dysfunction. Levodopa, a dopamine-precursor, is currently one of the most effective treatments for PD (Hornykiewicz, 2010). While the mechanism of action of levodopa is due to its ability to at least partially replete dopamine deficiency, few studies to date have focused on studying the effects of levodopa on brain physiology. Due to the lack of validated biomarkers for PD diagnosis or disease progression, current drug development in PD uses clinical outcome scales as measures of efficacy. This precludes quick assessment of new drugs in humans. Development of imaging biomarkers to assist in assessment of drug mechanism and target validation is crucial for development of effective new treatment regimens.

Neuroimaging techniques have been invaluable in providing the means to study drug effects in vivo, noninvasively. Fluorodeoxyglucose positron emission tomography (FDG-PET) has revealed in PD patients a network of brain regions with reduced glucose metabolism known as the PD Related Pattern (PDRP) (Eidelberg, 2009). The strength of its expression correlates with symptom severity, and is also reduced by levodopa treatment (Eidelberg, 2009). Similarly, 15O-PET has also shown that levodopa increases both resting cerebral blood flow (CBF) (Kobari et al., 1995) and motor activity (Feigin et al., 2002) in the putamen and thalamus of PD patients compared to healthy controls. Such findings have also been corroborated in blood oxygenation level dependent functional MRI (BOLD–fMRI) studies, which found that levodopa normalizes PD-related hypoactivity in regions such as putamen, M1 and supplementary motor area (SMA) (Buhmann et al., 2003; Tang & Eidelberg, 2010). Despite the high signal-to-noise ratio and ease of implementation, traditional BOLD techniques are non-quantitative and dependent on multiple physiological processes including blood flow, blood volume and oxygen consumption, and can thus be difficult to interpret in situations where more than one of these processes are affected.

Arterial spin labeling (ASL) (Detre et al., 1992) is another MRI technique that provides a quantitative measure of CBF by magnetically labeling blood spins without the need for contrast agent injection. In this regard, ASL is more similar to 15O-PET than traditional BOLD fMRI techniques as it provides a quantitative assessment of baseline physiology, albeit without radioactive exposure. Recent ASL studies have revealed similar hypoperfusion patterns in PD patients compared to controls as previously shown with PET and SPECT imaging (Eidelberg, 2009; Paschali et al., 2010; Song et al., 2015), including regions such as posterior parieto-occipital cortex, precuneus and cuneus and middle frontal gyri (Fernandez-Seara et al., 2012; Kamagata et al., 2011; Melzer et al., 2011). Furthermore, expression of the FDG-PET PDRP network is similarly increased in ASL CBF maps of PD patients compared to healthy controls and positively correlated to the expression in FDG-PET data of the same cohort (Ma et al., 2009; Teune et al., 2014). Since the ASL signal arises from magnetically labeled blood spins, which have a longitudinal relaxation time of 1.6 s at 3T, ASL can be easily repeated during a single scan session, making it well-suited for assessing the effects of a pharmacological challenge on brain activity (see Wang et al., 2011 for a review). The quantitative nature of ASL and its excellent reproducibility (Chen et al., 2011; Chen, Wang & Detre, 2011; Gevers et al., 2009; Hermes et al., 2007) also allow it to assess brain activity in the absence of tasks, making it especially attractive for studying patients by eliminating the potential confound of task performance. In this feasibility study, we aim to determine whether ASL can detect the effects of levodopa on resting CBF in a cohort of PD patients.

Methods

Subjects

A total of 16 patients (3 females, age 66.2 ± 8.7 years) with moderate PD severity (Hoehn and Yahr stage >2 (median 2.5) range, disease duration > mean (SD) 3 years) were recruited for the current study. One participant did not complete the postdrug session as the predrug images were corrupted by severe motion artifacts. Two other participants were removed from final analysis due to signal dropouts in the images which resulted in exaggerated CBF changes. Thus the final cohort consisted of 13 subjects. All participants were on a stable dose of carbidopa-levodopa therapy for at least 3 months prior to enrollment in the study. None of the participants had any visible head tremor as assessed by the clinicians on our team (DRG & PP) after dopaminergic medication was withheld approximately 12 h. This study was approved by the Northwestern Institutional Review Board (IRB ID: STU00049159) and written informed consent was obtained from all subjects prior to scanning. See Table 1 for a summary of the demographic details.

Table 1 Demographics, UPDRS-III ratings and Levodopa-equivalence (LDE) in mg.

Demographics (n=13)	
Age, years (mean ± s.d.)	65 ± 7	
Female: male	3:10	
Duration of disease (years)	8 ± 6	
LDEa, mg (mean ± s.d.)	270 ± 132	
L vs. R dominant symptoms	5:8	
Drug response (n=13)	
	Pre	Post	% improvement	
UPDRS-III total (mean ± s.d.)	19 ± 6	9 ± 6	52.0 ± 28.3	
Bradykinesia subscoreb	7 ± 4	4 ± 4	49.1 ± 38.8	
Tremor subscore	3 ± 2	1 ± 2	58.3 ± 43.3	
Rigidity subscore	5 ± 2	3 ± 2	46.8 ± 34.5	
Posture subscore	3 ± 1	1 ± 1	55.1 ± 39.8	
Notes.

a LDE, levodopa equivalents based on Tomlinson et al. (2010) for the morning dose of dopaminergic medications.

b See text for the formulas for the calculation of the UPDRS subscores.

Study protocol

Subjects were instructed to refrain from taking levodopa for at least 12 h prior to arrival at the MRI facility. Upon arrival, subjects underwent an assessment that included the Unified Parkinson’s Disease Rating Scale—motor (UPDRS-III) (Fahn S & Members of the UPDRS Development Committee, 1987) and then had an initial MR scanning session that consisted of the high resolution anatomical and ASL sequences. This session was designated the OFF or predrug state. Participants were then removed from the scanner and given an oral dose of carbidopa-levodopa approximately equivalent to 125% of their usual morning dose of dopaminergic medications calculated as levodopa equivalents (LDE) (Tomlinson et al., 2010). After 1 h, subjects were re-evaluated with the UPDRS-III and then had a second scanning session and ASL acquisition (ON or postdrug state). Although we had initially specified that subjects should show at least a 30% improvement in their UPDRS-III motor score for inclusion in the analysis, because of the small number of subjects we ended up retaining the 3 subjects with less than a 30% improvement (10%, 19% and 25%).

UPDRS-III subscores were calculated as follows for each session: (1) bradykinesia: sum of scores for finger tapping, hand movements, pronation-supination movements, leg agility and body bradykinesia; (2) rigidity: sum of rigidity scores for upper and lower extremities; (3) tremor: sum of rest and action tremor scores for upper and lower extremities; and (4) posture: sum of scores for posture, gait and postural stability. Lateralized scores were summed to generate a single number for each subscore for each subject.

All data were collected on a Siemens TIM Trio 3T whole-body scanner (Erlangen, Germany), equipped with a 12-channel receive-only head coil. A high resolution T1-weighted anatomical image was acquired using an MPRAGE sequence (TR/TE 2300/2.98 ms, TI = 900 ms, flip angle = 9°, GRAPPA acceleration factor = 2, matrix 256 × 256, voxel size 1 × 1 × 1 mm3, 176 slices). Cerebral blood flow maps were collected in both scan sessions using pseudo-continuous arterial spin labeling (pCASL) (Dai et al., 2008). ASL images were acquired with gradient-echo echo-planar imaging (EPI) with a field-of-view of 22 cm, and in-plane resolution of 3.5 × 3.5 mm2. Twenty, 6 mm axial slices with a 2 mm gap were prescribed to cover the entire brain including the cerebellum. Based on previous ASL studies on PD patients, labeling duration and post-labeling delay were both set to 1.5 s (Fernandez-Seara et al., 2012; Melzer et al., 2011). Forty pairs of tag and control images were acquired in an interleaved fashion, resulting in a scan time of 5.5 min.

Data processing

All images were processed using SPM8 (Wellcome Trust Centre for Neuroimaging, London, UK) and in-house developed scripts written in Matlab R2010b (Mathworks, Natick, MA, USA). The raw EPI images from both ASL scans were first co-registered to the high resolution anatomical image collected in session 1 and then motion-corrected by aligning all images to the first image of the session using a rigid-body, six parameter transformation. Pairwise subtraction images were computed for each control and tag pair, and outliers were excluded from the computation of the mean perfusion weighted image based on previously published guidelines (Wang et al., 2008). An image was labeled as outlier if it fit any of the following criteria: (a) rotation greater than 0.8°, (b) translation greater than 0.8 mm between the control and tag pair, (c) mean signal more than two standard deviations from the mean signal of the time-series or (d) mean background noise more than two standard deviations from the mean background noise of the time-series. On average, 7 ± 3 and 8 ± 4 pairs were labeled as outliers in the pre- and post-drug sessions, respectively, using the above criteria, indicating no significant difference in data quality between sessions. The mean perfusion weighted image was converted to CBF in physiological units using a previously published model (Wang et al., 2008). Mean global gray matter (GM) CBF values for baseline and post-drug conditions were computed from subject-specific masks encompassing voxels containing at least 50% GM, generated by segmenting the T1 image with the VBM8 toolbox (http://dbm.neuro.uni-jena.de/vbm) (Ashburner & Friston, 2000) and compared using a paired t-test. The low resolution CBF images were then spatially normalized to MNI space by applying the transformation matrices obtained from normalizing the high resolution anatomical images to a symmetrical template using VBM8.

To determine the most affected hemisphere across subjects, a motor impairment laterality index was calculated using the formula (R − L)/(R + L), where R and L represent the sum of the right and left limb UPDRS-III scores in the OFF drug condition respectively. Four subjects had a laterality index of less than 0, indicative of left-predominant symptoms, which would affect the right hemisphere. One subject had a laterality index of 0, with reported onset of symptoms on the left side. Since prior animal and human studies have demonstrated a close relationship between symptom laterality and asymmetry of biochemical changes in the brain (Eidelberg et al., 1990; Guttman et al., 1988; Huang et al., 2001; Morrish, Sawle & Brooks, 1995; Prakash et al., 2012; Rinne et al., 1993; Yang et al., 2007), and CBF is related to synaptic activity (Lauritzen, 2001a) that is also affected by underlying biochemical activity, images from these five subjects were flipped such that the most affected hemisphere for all subjects was on the left side in MNI space, thereby minimizing pathological variability due to symptom laterality. This is similar to the approach employed by another study that designated regions of interest as contra- or ipsilateral depending on the side of symptoms (Schuff et al., 2015). All spatially normalized CBF maps were smoothed with an 8 mm full-width at half maximum Gaussian kernel.

Statistical comparisons

Threshold-free cluster enhancement (TFCE) (Smith & Nichols, 2009) and nonparametric permutation testing (Nichols & Holmes, 2002) with 2,000 permutations were used for statistical analysis as these techniques have previously been shown to have superior sensitivity than traditional parametric statistics, and are particularly well-suited for studies with low degrees of freedom. Calculations were performed using the TFCE toolbox for SPM8. All results were thresholded at p < 0.05, with family-wise error (FWE) correction for multiple comparisons.

The following voxelwise comparisons were performed to further analyze the data.

1. Paired t-test: predrug vs. postdrug CBF

The predrug and postdrug CBF maps of all subjects were entered into a voxelwise paired t-test to explore the effects of levodopa on CBF. Two separate t-tests were performed on both the quantitative CBF maps and CBF maps proportionally scaled to 50 in order to minimize variance due to individual differences in CBF. Statistical analysis was limited to voxels with a raw signal to noise ratio (SNR) of at least 1 in both sessions for all 13 subjects, where raw SNR was calculated as the mean signal over time divided by the standard deviation of signal in a ghost-free background region.

2. Regression: drug-induced change in CBF vs. change in UPDRS III

To explore the relationship between levodopa-induced CBF changes and changes in motor function, normalized CBF difference maps were calculated by subtracting the predrug maps from the postdrug maps and normalizing by the predrug maps. Changes in UPDRS-III total and subscores were computed by subtracting the postdrug scores from the predrug scores so that a larger positive number represents greater motor improvement. Voxelwise multiple regression was used to regress the CBF difference maps against changes in UPDRS-III total and subscores.

3. Regression: drug-induced CBF change vs. predrug UPDRS-III

A similar approach was used to determine the relationship between drug-induced normalized CBF changes and baseline symptom severity by regressing the CBF difference maps against predrug UPDRS-III total and subscores.

4. Regression: drug-induced CBF changes vs. LDE

We also examined whether the induced CBF changes were related to the amount of levodopa administered with a multiple regression analysis of the drug-induced normalized CBF change maps using LDE values as the covariate of interest.

Results

As listed in Table 1, participants had an average 52.0 ± 28.3% improvement in their UPDRS-III total score. The minimum improvement for any subject was 10%. Also noted in Table 1, the average improvement in subscores was 49.1 ± 38.8% for bradykinesia, 46.8 ± 34.5% for rigidity, 58.3 ± 43.3% for tremor and 55.1 ± 39.8% for posture/gait.

1. Paired t-test: predrug vs. postdrug CBF

The mean global CBF extracted from voxels containing at least 50% GM was 46.2 ± 7.1 ml/100 g/min before levodopa and 47.6 ± 7.9 ml/100 g/min after levodopa, the difference was not statistically significant (t12 = − 0.78, p = 0.22).

In the voxel-wise analysis, no significant CBF changes were detected with the unscaled, raw, quantitative CBF maps, which was as expected since the inter-subject differences in global CBF far outweighed the drug-induced changes. Thus, only results from the proportionally scaled images are reported below. Figure 1 shows the clusters with significant CBF increases after levodopa, thresholded at p < 0.05 FWE, overlaid onto a high-resolution anatomical image. The image is oriented so that the “more affected” side of the brain (subject brains flipped as described above) is on the left side of the image. The colorbar represents the range of log scaled p-values. No levodopa-induced CBF decreases were detected. A summary of clusters, including MNI coordinates, FWE p-values, and approximate anatomical localization identified using the automatic anatomic labeling (AAL) toolbox (Tzourio-Mazoyer et al., 2002) for SPM are listed in Table 2. Mean CBF change extracted from a 5 mm sphere centered on the peak voxel of each cluster is also shown in Table 2. Levodopa induced CBF increases in the precuneus/cuneus, dorsal midbrain, less affected supramarginal gyrus, bilateral pre- and post-central gyri, less affected putamen/globus pallidus, more affected inferior frontal pars triangularis and opercularis, and more affected inferior parietal areas. Three clusters located in the white matter were also detected. However, given the controversy of white matter perfusion with ASL at 3T (Van Osch et al., 2009), we choose to not discuss these clusters further. The range of CBF changes in gray matter was between 27% and 74%.

Figure 1 Results of voxel-wise paired t-test between pre- and post-drug CBF maps thresholded at p < 0.05, FWE corrected for multiple comparisons, overlaid onto a representative subject’s spatially normalized anatomical image.

Images are oriented so that the “more affected” side of the brain is on the left side of the image. Orange represents CBF increases. No CBF decreases were detected at the statistical threshold used.

Table 2 List of clusters with significant CBF increases after levodopa administration.

MNI coordinates of the peak voxel are given in mm, as well as the number of voxels in each cluster and FWE p-value. Mean percent CBF changes were calculated from a 5 mm radius sphere centered at the peak voxel.

x, y, z, mm	No. voxels	FWE p-value	label	% CBF change	
0, −70, 33	2,695	0.016	Cuneus, precuneus	33.4	
56, −24, 34	3,046	0.020	SupraMarginal (LA), Frontal_Inf_Oper (LA), Postcentral (LA), Precentral (LA)	26.9	
4, −33 −9	544	0.027	Midbrain	73.6	
20, −6, −3	1,150	0.029	Pallidum (LA), putamen (LA)	44.0	
−60, 11, 9	461	0.030	Frontal_Inf_Oper (MA)	50.6	
10, −64, −5	863	0.030	Lingual (LA)	33.0	
−48, 8, 39	428	0.031	Precentral (MA)	31.0	
−34, −46, 60	314	0.043	Parietal_Inf (MA)	61.1	
34, −79, 9	108	0.043	Occipital_Mid (LA)	40.9	
−40, −18, 30	146	0.044	Postcentral (MA)	47.1	
38, −69, 15	77	0.044	White matter	47.8	
−40, −57, −23	108	0.045	Cerebelum_6 (LA)	36.2	
40, −27, −6	57	0.045	White matter	32.1	
−27, −85, 7	59	0.045	Occipital_Mid (MA)	47.3	
−58, −45, 39	77	0.047	Parietal_Inf (MA)	30.1	
−8, −28, −14	2	0.048	Midbrain	62.5	
38, −39, 6	26	0.049	White matter	85.8	
−50, 30, 25	5	0.050	Frontal_Inf_Tri (MA)	41.8	
Notes.

MA more affected

LA less affected

Note that left cerebellum was designated as less affected because motor fibers decussate in the medulla oblongata.

2. Regression: drug-induced change in CBF vs. change in UPDRS III

Only the change in bradykinesia UPDRS-III subscore had a significant negative correlation with drug-induced changes in CBF. This cluster, located in the left middle cingulate, is shown in Fig. 2A. The coordinates of the peak voxel in this cluster, as well as cluster size and FWE p-value are shown in Table 3.

Table 3 List of clusters with significant correlations between normalized ΔCBF and predrug or change in UPDRS scores.

MNI coordinates of the peak voxel are given in mm, as well as the number of voxels in each cluster and FWE p-value.

	x, y, z, mm	No. voxels	FWE p-value	Label	
ΔUPDRS					
Bradykinesia (negative)	−4, 3, 43	88	0.043	Cingulum_Mid (MA)	
Pre-drug UPDRS					
Rigidity (negative)	38, −57, 54	1,997	0.009	Angular (LA)	
	48, −12, 30	987	0.010	Postcentral (LA)	
	58, −27, 12	4,617	0.045	Temporal_Sup (LA)	
	−27, −60, 61	2,547	0.019	Parietal_Sup (MA)	
	−46, −48, 52	457	0.032	Parietal_Inf (MA)	
	2, −21, 72	6	0.044	Supp_Motor_Area (LA)	
	−34, −24, 49	106	0.045	Postcentral (MA)	
Total (negative)	−39, −9, 60	134	0.043	Precentral (MA)	
Notes.

MA more affected

LA less affected

3. Regression: drug-induced CBF change vs. predrug UPDRS-III

Clusters with significant correlations between change in CBF and predrug UPDRS-III scores are shown in Figs. 2B and 2C. Details on the clusters are summarized in Table 3. To demonstrate that the results were not driven by any outliers, scatterplots of ΔCBF, extracted from significant clusters and the corresponding UPDRS scores are shown in Fig. 3. Regression lines and adjusted R2 values are also shown on the plots, and are intended to demonstrate the strength of the regression in these clusters, not as an independent analysis of the data (i.e., circular analysis (Kriegeskorte et al., 2010)). The total UPDRS-III and rigidity subscores were significantly correlated with change in CBF. Total UPDRS-III score was negatively correlated to CBF changes in the more affected precentral gyrus. The rigidity subscore was negatively correlated to CBF changes in multiple areas including bilateral postcentral gyri, supplementary motor area and parietal and temporal regions.

Figure 2 Clusters with significant correlation between normalized changes in CBF and UPDRS-III subscores.

Clusters with significant correlation between (A) change in CBF and change in bradykinesia subscore, (B) change in CBF and predrug rigidity subscore and (C) change in CBF and predrug total UPDRS-III score. Blue represents negative correlations, where larger improvements were associated with smaller CBF changes. No positive correlations were detected at the statistical threshold used.

Figure 3 Scatterplots of normalized changes in CBF and UPDRS-III subscores.

Scatterplots of normalized change in CBF extracted from clusters with significant correlations to predrug and change in UPDRS-III total and subscores. Regression lines and adjusted R2 values are also shown.

4. Regression: drug-induced CBF changes with LDE

No significant correlation between drug-induced CBF changes and LDE were detected at the statistical threshold used.

Discussion

Drug induced CBF change (pre vs. post)

In this study, we used pCASL to measure CBF in PD patients before and after an oral dose of levodopa equivalent to 125% of their usual morning dosage of PD medications. The medication significantly improved motor functioning as expected; however, there was no significant change in global CBF. This finding is in agreement with several earlier 15O-PET studies investigating the effects of levodopa on CBF in both human PD patients and nonhuman primates (Hershey et al., 2003; Hershey et al., 2000; Hershey et al., 1998). The combination of the significant motor symptom benefits observed in our subjects and the lack of change in global CBF suggest that the levodopa dose administered in this study had minimal peripheral effects and had reached the intended target areas of the central nervous system, which were revealed by our voxel-based analysis.

Voxelwise paired t-test analysis between the proportionally scaled pre- and post-drug CBF maps revealed CBF increases after oral levodopa administration in several regions of the motor network including dorsal midbrain, (less affected) putamen/globus pallidus, and bilateral pre- and postcentral gyri. To further investigate the lack of bilateral results in the putamen/globus pallidi, we extracted putaminal CBF changes with hand-drawn posterior putamen masks. The left and right drug-induced putaminal CBF increases were 13% and 19% respectively, demonstrating bilateral putaminal response to levodopa. The lack of bilateral statistical significance is likely due to a combination of lower subcortical CBF and significant partial volume effects for these small structures. Our findings appear to contradict the negative findings from a similar ASL study which used pulsed ASL to study the effects of a steady infusion of levodopa in 12 PD patients (Stewart et al., 2014). The exact reason for this discrepancy is unknown. Since we did not measure blood concentrations of levodopa in the current study, it is unclear whether the dosage used is comparable between the two studies. Furthermore, the higher sensitivity of the TFCE technique for statistical analysis used in the current study may account for the different findings.

In addition to affecting regions within the nigrostriatal pathway, levodopa also induced CBF increases in the precuneus/cuneus, lingual, occipital and inferior parietal areas. While these areas are not traditionally associated with movement control, prior studies have reported perfusion deficits in similar regions for PD patients compared to healthy controls (Fernandez-Seara et al., 2012; Melzer et al., 2011). Such findings support the idea of PD as a multifaceted disease, as even regions extrinsic to the nigrostriatal network show significant involvement in PD. Indeed, both occipital cortex (Koh et al., 2013; Tessitore et al., 2012b) and precuneus (Melzer et al., 2011; Tang & Eidelberg, 2010; Tang et al., 2010; Tessitore et al., 2012a) have shown alterations in functional activation and structural integrity in Parkinson’s disease. The response of these regions to levodopa therefore may be mediated via their connections with other regions within the nigrostriatal network.

Correlation between drug-induced CBF change (pre vs. post), change in UPDRS-III scores and predrug UPDRS-III scores

In order to assess the regions of the brain that are associated with symptomatic improvement after levodopa administration, we correlated drug-induced CBF change to the change in UPDRS-III total and subscores. Only the change in bradykinesia subscore was significantly correlated with drug-induced CBF change, in a cluster located in the more affected mid-cingulate cortex, near the posterior end of Brodmann Area 24. The direction of the correlation was negative. Larger improvements are associated with smaller CBF changes. Both functional and diffusion tractography studies have demonstrated that the cingulate cortex can be segregated into functionally distinct regions (Vogt, Berger & Derbyshire, 2003). A comparison between the coordinates of this cluster with previous studies appears to categorize this cluster in the cingulate motor area (CMA), which has extensive functional connections to sensory-motor areas including the primary motor cortex and spinal cord (Torta & Cauda, 2011) and is often activated during motor tasks (Beckmann, Johansen-Berg & Rushworth, 2009). While the current finding appears in agreement with previous reports regarding the involvement of CMA in motor control, further research is necessary to elucidate why the CMA is associated with bradykinesia symptom improvement and not other parkinsonian symptoms.

We also performed a similar correlation analysis between CBF change and predrug UPDRS-III total and subscores to investigate whether disease severity affects the drug-induced CBF change. In this analysis, both total and rigidity UPDRS-III subscores were significantly correlated with CBF change. Specifically, the total UPDRS score was negatively correlated to CBF change in the more affected precentral gyrus. The rigidity subscore, on the other hand, revealed a widespread network of negative correlations, including bilateral postcentral gyri, superior motor area and parietal areas. The negative correlation indicates that more severe clinical symptoms are associated with smaller CBF changes. This relationship is consistent with the expectation that more severe clinical symptoms are likely associated with more disrupted neurocircuitry, resulting in a more blunted drug response.

Although our results are in general agreement with findings in PD using other modalities, it is important to emphasize that an assessment of regional CBF changes only partially addresses questions regarding underlying physiological processes. One ambiguity about CBF is that it is more tightly coupled to synaptic activity than spiking activity (Jueptner & Weiller, 1995; Lauritzen, 2001b), but it does not distinguish between excitatory or inhibitory input, making it difficult to determine downstream effects within a given network of regions. Also, an increase in CBF may not always represent an increase in local glucose metabolism or neural activity, as the typical tight coupling between CBF and metabolism may be disrupted in disease states. For example, concurrent CBF and glucose metabolism PET studies have shown in both rats and humans that levodopa induces a significant dissociation between CBF and glucose metabolism within the PD motor-related network, and that this dissociation is especially prominent in patients with levodopa-induced dyskinesia (Hirano et al., 2008; Ohlin et al., 2012). Future studies using multimodal techniques may help elucidate how functional hyperemia is affected.

This study was limited by a small sample size, with no control for placebo effects, and not all participants had a robust response to levodopa. As we did not include healthy, age-matched controls in this study, we are unable to determine whether the levodopa-induced CBF changes are specific to PD pathology. Correlation analysis between levodopa-induced CBF change and UPDRS motor scores suggest that the CBF response to levodopa may be different for patients on opposite ends of the severity spectrum. This remains to be demonstrated in a larger group of patients. Another limitation with the current study is the spatial resolution of the ASL scan, which makes it difficult to resolve substructures in several of the brain regions implicated in PD pathology, including the globus pallidus and substantia nigra, without partial volume effects. This will be exacerbated when comparing patients to healthy controls where different degrees of atrophy will also lead to variability in regional CBF. Higher resolution ASL techniques, combined with partial volume correction, would provide a more accurate assessment of local CBF changes. Finally, this study examined subjects with moderately advanced PD in the OFF state (12 h post last dose) and then the ON state. While this is an accepted method of examining medications ON/OFF effect, the OFF state does not reflect the true biological OFF due to a long duration effect of PD medications (Hershey et al., 2003). Ideally,drug-naïve PD patients should be imaged at baseline and followed longitudinally throughout their treatment for a better understanding of the underlying mechanism.

Conclusion

We used ASL, a noninvasive MRI technique for measuring CBF, to explore the effects of an acute oral administration of levodopa on resting CBF in PD patients. Levodopa induced CBF increases in many regions of the brain, including major components of the nigrostriatal pathway, suggesting a combination of direct vascular and functional effects. Correlation analysis demonstrates a possible interaction between disease severity and CBF response to acute drug challenge. These results suggest the feasibility of ASL as a probe for the physiological effects of drugs in PD, which could not only further our understanding of current drug mechanisms, but may also be useful for designing new treatments, following validation in larger studies.

Abbreviations

PD Parkinson’s disease

FDG-PET Fluorodeoxyglucose positron emission tomography

PDRP PD related pattern

CBF cerebral blood flow

BOLD-fMRI blood oxygenation level dependent functional MRI

SMA supplementary motor area

ASL arterial spin labeling

UPDRS Unified Parkinson’s Disease Rating Scale

LDE levodopa equivalent

pCASL pseudo-continuous arterial spin labeling

SPM statistical parametric mapping

SNR signal to noise ratio

CMRO2 cerebral metabolic rate of oxygen

Additional Information and Declarations

Competing Interests

Author Contributions

Human Ethics

Data Availability

The authors declare there are no competing interests. Darren R. Gitelman is an Academic Editor for PeerJ.

Yufen Chen performed the experiments, analyzed the data, wrote the paper, prepared figures and/or tables.

Peter Pressman performed the experiments, reviewed drafts of the paper.

Tanya Simuni contributed reagents/materials/analysis tools, reviewed drafts of the paper.

Todd B. Parrish conceived and designed the experiments, contributed reagents/materials/analysis tools, reviewed drafts of the paper.

Darren R. Gitelman conceived and designed the experiments, performed the experiments, reviewed drafts of the paper.

The following information was supplied relating to ethical approvals (i.e.,, approving body and any reference numbers):

Northwestern University IRB ID#: STU00049159.

The following information was supplied regarding data availability:

Figshare:

http://figshare.com/s/b022c95e743b11e59a1406ec4bbcf141

http://figshare.com/s/286cd544743c11e580be06ec4bbcf141

http://figshare.com/s/6568844e743b11e580be06ec4bbcf141.

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
