# Peer review of "Effects of acute levodopa challenge on resting cerebral blood flow in Parkinson’s Disease patients assessed using pseudo-continuous arterial spin labeling"

_PeerJ, doi:10.7717/peerj.1381_

## Round 0.1 · original submission · Major Revisions

The expert reviewers appreciated the potential value of the study but were skeptical of the results given the sample size and some choices about reporting. One recommended rejection and the other major revisions. On the whole, given the aim of PeerJ and its editorial criteria, I believe it may be possible to amend the analysis and discussion to be acceptable for publication.

I agree with the need to limit speculation about potential findings that do not pass correction for multiple correction. The initial threshold of p<.001 is a reasonable option, but the FWE correction acknowledged in Table 2 should be given more weight. I agree that including the other entries in this table, but excluding clusters with <50 voxels, does appear arbitrary.

Nevertheless, at least one region was statistically significant, and its peak is in motor cortex, so although the implications of the limited sample size must be acknowledged appropriately, I don't think it is too small for the results to be published, especially in a context such as PeerJ where the study data are made available.

Your discussion of whether levodopa disrupts metabolic-flow coupling was welcome. The issue is more nuanced. Ref. 25, Hirano et al, is widely known, but the data they report was not quantitative, so their conclusions about disrupted coupling are not valid, or at least go far beyond their data. In addition to refs. 23-24, the following reference also supports maintenance of coupling with levodopa: PubMed PMID: 11085899. On the other hand, these 3 Hershey et al refs, which provide the most direct support for validity of rCBF measurements of levodopa's effect, used an aggressive carbidopa dosing strategy that may have blocked peripheral dopamine production more effectively than the dosing strategy you used here. This is pertinent because, as you know, i.v. dopamine affects blood flow directly. Still, given carbidopa's half-life and your patient population, some carbidopa may yet have been present in blood when you gave the CD-LD. If you measured blood levels of either, they would strengthen the report. Consider adding these points to your discussion.

The discussion would be enriched by citing two other studies. One is another levodopa challenge ASL CBF study in PD (http://www.jneurosci.org/content/30/48/16284 , reanalyzed in https://peerj.com/articles/687/, the last paragraph before Discussion). The other line of work comes from 2DG studies of levodopa's effect on regional metabolism in healthy and hemiparkinsonian rodents by J. M. Trugman and colleagues, e.g. PubMed PMID numbers 8836523, 8539420, 1829645, or 3527339.

In your response letter, please respond to each of the reviewers' comments and the comments above.

Reviewer 1 ·

Basic reporting

A few minor issues:
1) Threshold for excluding subjects for head motion as PD subjects are known to have difficulty staying still
2) What is definition of lack of detectable head tremor
3) Ref 14 is 2011
4) There are several recent papers on ASL in PD that should be included.

Experimental design

As the authors acknowledged, this is a pilot study without placebo control and healthy control subjects. The flipping of L/R for dominant symptom seems lacking support from literature.
Did the authors look at any correlation with the dose of levodopa? The dose should be included in Table 1. The authors reported increased and decreased CBF responses, some of which may be related to withdraw effects.

Validity of the findings

The authors used an arbitrary threshold of p<0.001 with a cluster size of 50 pixels and cited 2 AD papers. It is understandable that ASL has lower SNR compared to BOLD fMRI, thus may not pass rigorous correction of multiple comparisons. However, there are several more objective methods the authors should consider e.g. permutation based correction (alphasim), small volume correction based on a priori ROIs.

Additional comments

This is a pilot study using pCASL to study CBF response to acute oral administration of levodopa in a small cohort of moderate PD subjects. The results are interesting and may be of value for future studies. However, the study design has many potential confounding factors, so are analyses (see detailed comments in each category). I recommend major revision.

·

Basic reporting

No comments

Experimental design

The authors have presented an arterial spin labeling study of medication effects in a group of patients with moderate Parkinson’s disease. There are a number of concerns with the design and its presentation.

1. The number of patients is very low. A group size of 13 is smaller than would be expected in any pharmacological imaging study with sufficient power to detect differences that the field would reasonably expect could be replicated. This is a major concern. There is sufficient data from prior PET studies to conduct a power analysis for a small number of regions, which would have been informative regarding the power of the present study.
2. The use of the withdrawal paradigm is widespread and understandably pragmatic. Nonetheless, the effects observed could be related to withdrawal (over 12 hours) or the medication effect or a combination of both. This limitation must be acknowledged. Without a longer withdrawal period the degree of withdrawal related contribution to the observed effects is unknown.
3. The authors themselves acknowledge that there was no control or placebo group. In the absence of such groups the purpose of the study must be questioned. The authors state that ‘we aim to test whether ASL can detect the effects of levodopa on resting CBF in a cohort of PD patients.’ If the authors wish to attribute the changes observed to levodopa then the fixed order of scanning, the knowledge that levodopa precedes the second scan and different times of day will all contribute to the variance in the results.
4. The ASL sequence, while reasonably standard includes quite a short post-labeling delay of only 1.5s thus allowing for vascular effects to contribute to the maps. The recent paper by Alsop et al. (doi: 10.1002/mrm.25197) should be cited here.

Validity of the findings

There are a number of concerns with the statistics and data presentation.

1. A major issue is the choice of statistical thresholds. As the authors acknowledge, only one cluster actually survived multiple comparisons correction. The threshold chosen for reporting thus appears arbitrary (p<0.001 and 50 voxels). A formal multiple comparisons correction scheme must be used. For ASL studies many papers use P<0.01 and cluster correction, although a method such as TFCE might be more appropriate.
2. The use of global correction is acceptable but can result in unclear findings, including a change, which is not present in the absolute data, but comes out as a relative change. For this reason I would encourage the authors to mention in the text which of their findings were also present without global correction.
3. It is not appropriate to take significant findings in imaging and plot their effect sizes as it is recognized that these effect sizes will be inflated. This is clearly the case for the regression plots. That is the statistical inference and effect size calculation was completed for the whole brain analysis. The additional presentation of R squared values not acceptable, although the scatter plots without regression is just fine (Kriegeskorte, Lindquist et al, JCBFM 2010).
4. The labeling of regions is not consistent. First, the changes due to l-dopa withdrawal/administration in Figure 1 are tabulated in Figure 2, but the findings subsequently are not tabulated and it is not clear why the authors made this decision. A larger table including all peaks would be more useful as currently the reader is limited to the slices chosen by the authors to see the regression results. For the region labeled SMA and displayed on Figure 2 it is not clear that this is unequivocally SMA. It could easily be labeled cingulate gyrus and certainly falls within the bounds of BA24. This is important and may change any interpretation (if this result remains significant).

Additional comments

Overall, the aims of this study seem useful and the use of ASL is reasonably justified. However, the implementation falls short of really addressing the questions well, due to the lack of controls, the low numbers and poor statistics.

---

## Round 0.2 · accepted · Accept

The authors have appropriately addressed my and the reviewers' concerns.

It is not relevant here because gCBF increased (slightly), so artifactual regional changes would have been decreases, and you did not detect any regional decreases. Nevertheless, purely for future reference, I think reviewer 2's intended point about global changes was slightly different and could be better addressed differently. See for instance figures 4 and 5 from a pharmacological challenge PET study, http://www.ncbi.nlm.nih.gov/pmc/articles/PMC139278/figure/f4/ and http://www.ncbi.nlm.nih.gov/pmc/articles/PMC139278/figure/f5/ , and the accompanying text.